# Effect of Fenugreek (*Trigonella foenum-graecum* L.) Seed Extracts on the Structure of Myofibrillar Protein Oxidation in Duck Meat

**DOI:** 10.3390/foods12244482

**Published:** 2023-12-14

**Authors:** Mingyue Chen, Qingmei Pan, Binbin Wu, Hongxun Wang, Yang Yi, Wei Xu, Danjun Guo

**Affiliations:** 1School of Food Science and Engineering, Wuhan Polytechnic University, Wuhan 430023, China; cmy6604@163.com (M.C.); wu826265817@163.com (B.W.); wanghongxunhust@163.com (H.W.); yiy86@whpu.edu.cn (Y.Y.); missguodj@163.com (D.G.); 2Key Laboratory for Deep Processing of Major Grain and Oil, Ministry of Education, Wuhan Polytechnic University, Wuhan 430023, China; 3Hubei Key Laboratory for Processing and Transformation of Agricultural Products, Wuhan 430023, China; 4Hongan County Public Inspection and Testing Center, Hongan 438400, China; panpanqingmei@163.com

**Keywords:** fenugreek seed, duck meat, myofibrillar protein, antioxidation

## Abstract

The effect of fenugreek (*Trigonella foenum-graecum* L.) seed extracts (FSEs) on the structure of duck myofibrillar protein (MP) oxidation was researched via particle size, zeta potential, Fourier transform infrared (FTIR), fluorescence spectroscopy, SDS-PAGE, and scanning electron microscopy (SEM) in the Fenton oxidation system. FSE (0.3 mg/mL) could scavenge 58.79% of the hydroxyl radical and possessed good antioxidation. FSE could retard the oxidation of MP, and the carbonyl formation and total sulfhydryl loss of MP decreased by 42.00% and 105.94%, respectively, after 4.67% of FSE treatment. SDS-PAGE results showed that 0.67% and 2.67% of FSE decreased the strength of the myosin heavy chain (MHC) and actin bands of the oxidized MP, respectively. The FSE changed the secondary structures of the MP and promoted the unfolding of the MP structure and the transformation from α-helix to β-turn. When treated with 0.67% of FSE, the hydrophobicity of the MP declined by 26.14%, and solubility was improved by 37.21% compared with the oxidation group. After 0.67% of FSE treatment, the particle size and zeta potential of the MP returned to the level of the blank group. Scanning electron microscopy revealed that FSE improved the apparent morphology of the MP. Overall, FSE had positive effects on the antioxidation of the duck MP, and it could improve the structure and characteristics of the MP. It is hoped that FSE could be considered as a natural antioxidant to retard the oxidation of the MP in meat products.

## 1. Introduction

Duck products are popular in Asian countries for their nutrition, accounting for 84.2% of the world’s total duck meat production [1]. Peking roast duck, Nanjing salted duck, marinated duck, and spicy salted duck are common duck products in Asian countries. Meat is an important source for humans to supplement protein. The myofibrillar protein (MP) is the most abundant muscle protein in meat, accounting for 55–60% of muscle protein [2]. The MP can impact meat products with factors such as tenderness, texture, and flavor [3,4]. The MP was easily oxidized during the thermal processing. It was reported that the divulgence of meat to oxygen would produce reactive oxygen species (ROS) and then induce protein oxidation [5]. Protein oxidation would change the structure of the MP and lead to protein aggregation and degradation, thus affecting the sensory quality and nutritional values of meat products [6,7,8]. It was found that protein oxidation caused by hypochlorous acid increased the particle size and carbonyl content of the MP [9]. The latest study found that the contents of carbonyl and total sulfhydryl of the duck MP significantly increased under a simulated high-temperature environment, and the surface hydrophobicity was also enhanced by high temperature [10]. Severe oxidation of proteins increases a number of conformational changes, such as random coils [11]. It was reported that the oxidation of beef myofibrillar led to a decrease in the water-holding capacity and tenderness in beef meat [12]. Therefore, deceleration of protein oxidation during the heat treatment is highly important to maintain the edible quality of meat products.

Antioxidants are normally added to meat processing to reduce the oxidative deterioration of meat and protein and extend the shelf life of meat products [13]. Synthetic antioxidants such as butylated hydroxy anisole (BHA, the maximum use level is 0.2 g/kg) and butylated hydroxytoluene (BHT, the maximum use level is 0.2 g/kg) are the most commonly used antioxidants, which have great toxicity and side effects on human health [14,15]. Numerous studies found that some plant extracts, such as native Chilean plants, yam saponin, citrus fruit (hesperidin), ginger, black currant (*Ribes nigrum* L.), and pomegranate peel extracts, have a significant antioxidant effect in the heating process of a meat product [16,17,18,19]. Among the numerous natural antioxidants, fenugreek seed has strong antioxidant potential and high safety [20]. 

Fenugreek (*Trigonella foenum-graecum* L.) seeds are the seeds of the annual plant fenugreek with a unique and rich caramel aroma [21] and are economic crops with rich resources [22]. It was reported that fenugreek seed extracts improved the oxidative stability of pork frankfurters [23] and exhibited effective antimicrobial properties and high antioxidant activity in lamb and beef [24]. In our previous study, the fenugreek seed extracts (FSEs) obviously decreased the shearing force of duck muscle and improved the tenderness of duck meat. However, the antioxidative effects of FSE on the MP in duck meat and the corresponding mechanism are not clear at present. Therefore, the mechanism of FSE–MP interaction and the antioxidant effect of FSE on the structure of the duck meat myofibrillar protein were investigated. The purpose of this study was to provide a natural plant extract for the oxidative regulation of meat products, and realize the high value utilization of FSE while maintaining the good sensory quality of meat products.

## 2. Materials and Methods

### 2.1. Materials and Reagents

Fenugreek seeds (dried mature seeds of fenugreek, family Leguminosae, originating from Ningxia Hui Autonomous Region Ningxia, China) were purchased from the Langqiao food franchise store (Wuhan, China). Frozen duck drumsticks (Pekin Duck) were purchased from Henan Huaying Agriculture Development Co. (Huaying, China). Bovine serum albumin was purchased from Wuhan Feiyang Biotechnology Co., Ltd. (Wuhan, China). L-ascorbic acid, glycine, Piperazine-1,4-bisethanesulfonic acid (PIPES), 5,5′-Dithiobis-(2-nitrobenzoic acid) (DTNB), bromophenol blue, Dinitrophenylhydrazene (DNPH), 1,1-Diphenyl-2-picrylhydrazyl radical 2,2-Diphenyl-1-(2,4,6-trinitrophenyl) hydrazyl (DPPH), hydrogen peroxide (H_2_O_2_, 30%), ferric chloride (FeCl_3_), hydrogen chloride (HCl), dicalcium chloride (KCl), magnesium chloride (MgCl), ferrous sulfate (FeSO_4_), (hydroxymethyl)aminomethane (Tris), EthyleneDiamine Tetraacetic Acid (EDTA), trichloroacetic acid, ethyl acetate-ethanol solution, guanidine hydrochloride solution, salicylic acid, β-mercaptoethanol, Phenylmethanesulfonyl fluoride, and phosphate-buffered saline (all of the analytical grade) were purchased from Sinopharm Group Chemical Reagent Co., Ltd. (Shanghai, China).

### 2.2. Extraction of the Duck Meat MP

The duck MP was prepared following a previously established method with necessary adjustments [25]. The raw material used in this study was peeled duck leg meat. The duck leg was thawed at 4 °C for 4 h using a HYC-326A refrigerator (Qingdao Haier Co., Ltd., Qingdao, China). Subsequently, 5 g of meat was pre-treated to clean out the connective tissue and added to 50 mL of a buffer solution (4 mmol/L EDTA-2Na, 3 mmol/L MgCl_2_, 1 mmol/L Phenylmethanesulfonyl fluoride, 150 mmol/L NaCl, 25 mmol/L KCl, and pH 6.5). The mixture was then diluted and homogenized using the XHF-DY high-speed disperser (Ningbo Xinzhi Biotechnology Co., Ltd., Ningbo, China) at 10,000 r/s for 10 s. During homogenization, physical cooling with ice was employed. The supernatant was separated via centrifugation (10,000 rpm, 10 min, and 4 °C), and the resulting precipitate was washed with a solution (5 mmol/L β-mercaptoethanol and 50 mmol/L KCl, pH 7). Lastly, the precipitate was extracted with a solution (5 mmol/L β-mercaptoethanol, 50 mmol/L KCl, and pH 7), homogenized at 10,000 r/s for 10 s, and the resulting white paste was obtained as the MP, and the MP was freeze-dried to a powdered state for subsequent experiments. The protein content was calculated using the Coomassie brilliant blue.

### 2.3. Extraction of FSE and Construction of the Fenton System

Based on previous studies conducted in our laboratory, the extraction of fenugreek seed extracts (FSEs) was carried out using the following procedures: Firstly, fenugreek seeds were dried at 65 °C for 8 h in an electric thermostatic drying oven (Shanghai Yiheng Technology Co., Shanghai, China), then crushed and sieved. The crushed fenugreek seeds and 70% ethanol were configured in a material–liquid ratio of 1:15, sonicated using the Ultrasonic Cleaner (Ningbo Scientz Biotechnology Co., Ningbo, China) for 1.5 h (140 W, 50 °C), and immersed for 8 h. The ethanol extract of fenugreek seeds was subsequently vacuum filtered and rotary evaporated (25 rpm and 55 °C) to remove ethanol and excess water. The concentrated ethanol extract of fenugreek seeds was freeze-dried for 72 h, resulting in the formation of FSE in the form of a freezing powder. 

The Fenton system was constructed according to a previously established method with minor modifications [26]. The Fenton system was constructed by adding FeCl_3_, ascorbic acid, and H_2_O_2_ solution [27]. The MP and FSE were diluted to the desired concentrations with a 15 mmol/L PIPES buffer (0.6 mol/L NaCl, pH 6.25). Each MP solution group contained the following components: 10 mg/mL MP, 0.01 mmol/L FeCl_3_, 0.1 mmol/L ascorbic acid, and 10 mmol/L H_2_O_2_. The mixture was agitated evenly and oxidized (12 h and 4 °C). The protein solution with 0.67%, 2.67%, and 4.67% (*w*/*w*) of FSE concentration was used as the FSE-L, FSE-M, and FSE-H of the experimental group, respectively. The protein solution with the Fenton oxidation system was used as the oxidation group (OX), while the protein solution containing EDTA-2Na served as the blank control (blank).

### 2.4. Hydroxyl Radicals Scavenging Capacity

The hydroxyl radicals scavenging capacity was measured using a previous method with appropriate modifications [28]. Briefly, 1 mL FeSO_4_ (2 mmol/L), 1 mL H_2_O_2_ (6 mmol/L), and 1 mL (6 mmol/L) salicylic acid were mixed and water bathed (37 °C for 15 min). A total of 7 mL of dilution water was added to 3 mL of the mixture and shaken well. The absorbance was measured at 510 nm using a UV–visible spectrophotometer (Younico Instrument Co., Ltd., Shanghai, China), and the measurement was recorded as A_0_. Similarly, the absorbency of 7 mL of the sample (at concentrations ranging from 0.05 to 0.35 mg/mL) mixed with the above solution was measured as A_x_. The hydroxyl radical scavenging rate was calculated using the following formula: (1)$$\mathrm{Hydroxyl}\;\mathrm{radical}\;\mathrm{scavenging}\;\mathrm{rate}\left(\%\right)=\frac{A_{0}-A_{x}}{A_{0}}\;\times \;100$$

### 2.5. Determining Carbonyl Content

The total carbonyl content was assessed using the method reported in prior work with a few minor adjustments [29]. A total of 2 mL aliquots of the duck meat MP solution were mixed with 2,4-dinitrophenylhydrazine (10 mmol/L and DNPH) in a 2 mol/L HCl solution at 25 °C for 1 h in dark. Subsequently, 2 mL of 20% trichloroacetic acid was combined with the above mixture. The supernatant was removed after centrifugation (4000 rpm, 10 min, and room temperature), and the precipitate was then washed with 1 mL of ethyl acetate–ethanol solution (1:1, *v*/*v*) to remove any unreacted DNPH. After the washes, the organic solvent was evaporated, and 6 mL of a 6 mol/L guanidine hydrochloride solution was added. The mixture was incubated (15 min, 37 °C) and then centrifuged (2000 rpm, 3 min, room temperature) to obtain the supernatant, then measured at 370 nm. The carbonyl concentration was determined using the molar absorptivity of 22,000 M^−1^ × cm^−1^ and represented as nmol/mg protein.

### 2.6. Determining Total Sulfhydryl Content

Sulfhydryl contents were measured using the reported means [19] with modifications. Firstly, 1 mL MP (2 mg/mL) was combined with 8 mL of a buffer (0.09 mol/L glycine, 4 mmol/L EDTA, 0.086 mol/L Tris, and pH 8), and the mixture was centrifuged (10,000 rpm, 15 min, room temperature) to obtain the supernatant. Secondly, 0.5 mL of Ellman’s reagent (10 mmol/L DTNB) was added to 4.5 mL of supernatant, and the mixture reacted (30 min, room temperature). The absorbance of the combined solution was determined at 412 nm using a UV–Vis spectrophotometer. The sulfhydryl content was computed using the sulfhydryl molecule’s absorption coefficient of 13,600 M^−1^ × cm^−1^, and the results were shown as nmol/mg. 

### 2.7. SDS-PAGE Electrophoresis

The SDS-PAGE gel electrophoresis measurement was measured using a reported method [30] with modifications. The mass concentration of the MP solution was prepared to 1 mg/mL in the blank, OX, FSE-L, FSE-M, and FSE-H groups, respectively. The 5% concentrate gel and 12% separator gel were prepared in advance. Experimental operations were performed on a JY04S-3C Gel Imaging System (Beijing Junyi Oriental Electrophoresis Equipment Co., Beijing, China) after the sample was prepared. A total of 15 μL of the sample was added to the wells of the prepared gel and energized with a Tricine SDS Dispensing Buffer (60 V, 30 min), and then the voltage was adjusted to 120 and continued to energize for 40 min. After electrophoresis, the protein gel was subjected to coloring, decolorization, and imaging to observe the cross-linking and degradation of MP in the five groups.

### 2.8. Fourier Transform Infrared (FTIR) Spectroscopy

The FTIR spectrometry of the MP was detected according to a reported method [31]. A total of 2 mg of freeze-dried MP samples and 200 mg of completely dried KBr were fully ground and mixed. The treated samples were scanned using a near-infrared spectrometer (N-500, HITACHI, Japan) from 400 to 4000 cm^−1^. The second derivative fitting of the Amide I Bands (1600 to 1700 cm^−1^) was detected using the software Peakfit v4.12, and changes in each secondary structure, including α-helix, β-sheet, β-turn, and random coil, were calculated using a reported method [32]. 

### 2.9. Fluorescence Spectroscopy of Endogenous Tryptophan

The measurement of fluorescence spectroscopy of endogenous tryptophan was referred to in the reported study [33]. The sample was diluted in a 15 mM/L PIPES buffer to a concentration of 0.4 mg/mL, and the samples were detected in a quartz cuvette with an optical range of 1 cm with excitation at 283 nm. The fluorescence spectroscopy of the sample was collected from 300 to 450 nm using a fluorescence spectrophotometer (F-7000 HITACHI) (Hitachi, Ltd., Tokyo, Japan).

### 2.10. Particle Size

The particle size of the MP was examined using the reported means [34] with slight modifications. The MP sample was diluted in PBS to a concentration of 1 mg/mL, and the heat-treated MP solution was centrifuged (10,000 rpm, 10 min, 4 °C). The particle sizes of the MP were detected using the particle electrophoresis instrument (Zetasizer Nano-ZS90, Malvern Instruments Ltd., Malvern, UK) at 25 °C.

### 2.11. Zeta Potential

The measurement of zeta potential was determined according to a previous method [35] with modifications. A total of 1 mg/mL MP samples were centrifuged (10,000 rpm, 10 min, 4 °C), and the supernatant was taken as the test sample. The zeta potential of the MP was measured using the particle electrophoresis instrument (Zetasizer Nano-ZS90, Malvern Instruments Ltd., UK) at 25 °C. 

### 2.12. Determining Surface Hydrophobicity

The surface hydrophobicity of the MP (the protein concentration of MP was 10 mg/mL) was measured according to the bromophenol blue binding method (BPB) [10] with slight modifications. A total of 40 μL of BPB (1 mg/mL) with a 1 mL MP suspension and a 1 mL pH 6.0 20 mmol/L phosphate buffer were regarded as the sample group and control group, respectively. The two groups were then shaken (10 min, 25 °C) and centrifuged (4000 rpm, 15min, room temperature). The absorbance of the supernatant was measured at 595 nm using a phosphate buffer as a reference. The BPB binding amount was used as the hydrophobicity index and calculated using the following formula:(2)$$\mathrm{BPB}\;\mathrm{binding}\;\mathrm{amount}\;\left(\mu g\right)=40\times \frac{{\mathrm{OD}}_{\mathrm{control}}-{\mathrm{OD}}_{\mathrm{sample}}}{{\mathrm{OD}}_{\mathrm{control}}}$$

### 2.13. Solubility

The solubility measurement was carried out according to the method of predecessors [18] with appropriate revision. The MP was digested, and the total protein concentration of the MP was determined according to the Kjeldahl nitrogen method. A total of 10 mg/mL of MP solution was homogenized (12,000 rpm, 1 min) and centrifuged (6000 rpm, 15 min, room temperature). The protein concentration of the MP supernatant was determined using the biuret method [36]. The protein solubility was calculated using the following formula: (3)$$S\mathrm{olubility}\;\left(\%\right)=\frac{P_{s}}{P_{T}}\times \;100$$
where P_S_ (mg/mL) is the protein concentration of the MP supernatant, and P_T_ (mg/mL) is the total protein concentration of the MP. 

### 2.14. Scanning Electron Microscope Analysis

The microstructure of the MP was determined using the JMS-6700F scanning electron microscope (JEOL Ltd., Tokyo, Japan), which was referred to as the previous method [37]. MP samples from the blank, TM, FSE-L, FSE-M, and FSE-H groups were freeze-dried for 48 h and then observed with the microscope. The graphs were collected at 500× and 1000× magnification.

### 2.15. Statistical Analysis

All experiments were set up in three parallel groups, and the results were expressed as mean ± standard deviation in this study. Experimental results were analyzed with an IBM SPSS 26 with a one-way analysis of variance (ANOVA) and Duncan’s multiple range test. When the *p*-values were less than 0.05, they were considered statistically significant, and Origin 2022 was used to draw the figures. 

## 3. Results and Discussion

### 3.1. Antioxidant Capacity of FSE

As Figure 1 shows, when the concentration of FSE varied from 0.04 to 0.12 mg/mL, the clearance rate of the hydroxyl radical (OH) increased from 38.85% to 58.79%. The results showed that FSE possessed superior hydroxyl radical-scavenging activity and oxidation resistance. It might be because flavonoids, polysaccharides, and other antioxidants were the main components of FSE [38]. Therefore, FSE could be used to prevent protein oxidation in duck meat.

### 3.2. The Effect of FSE on MP Oxidation

#### 3.2.1. Carbonyl and Total Sulfhydryl Contents in MP

Protein oxidation is an essential factor in meat quality deterioration, attracting increasing attention to meat products [39]. Formation of carbonyl and loss of sulfhydryl were common assessment methods to evaluate the degree of protein oxidation [40,41,42]. In Figure 2a, the carbonyl content is significantly decreased in all FSE groups compared with the OX group (*p* < 0.05), and there is no significant difference between the blank group and the FSE-H group (*p* > 0.05). It suggested that FSE could inhibit MP oxidation induced via the Fenton oxidation system in a concentration-dependent manner. Flavonoids and phenols in FSE might be good hydrogen donors with an excellent capacity to scavenge free radicals [41]; thus, the FSE could effectively inhibit the emergence of carbonyl. It was also reported that green tea and rosemary extract could effectively inhibit the formation of protein carbonyl in sausages derived from oxidatively stressed pork [43]. Similarly, mulberry (*Morus alba*) polyphenols showed lower carbonyl contents than the negative control in Cantonese sausages [44].

In Figure 2b, the total sulfhydryl content of MP in the OX group is decreased by 57.20% compared to the blank group. In the FSE groups, loss of the total sulfhydryl contents was significantly decreased compared with the OX group (*p* < 0.05). This result showed that FSE could reduce the loss of sulfhydryl groups induced by the Fenton oxidation system and inhibit the process of protein oxidation. The total sulfhydryl content in the FSE-L group even reached a higher value than that in the blank group (*p* < 0.05). However, with the increase in FSE concentration, the loss of sulfhydryl content increased. It might be due to the fact that the bioactive compounds in FSE (such as phenolic) interacted with sulfhydryl groups via the covalent interaction in the MP to form reaction products such as sulfhydryl–quinone, causing a significant decrease in the sulfhydryl levels of the MP [38,45]. A previous study also reported that a high dosage (1 mg/mL protein) of ginger, yam saponin, and hesperidin extracts could significantly decrease the sulfhydryl content of oxidative chicken myofibrillar proteins compared with the low-dosage (0.5 mg/mL protein) groups [16]. 

#### 3.2.2. SDS-PAGE Electrophoresis of MP

SDS-PAGE electrophoresis was used to explore the effect of FSE on oxidative MP, and the intensity of the protein bands is presented in Figure 3. In this study, the myosin heavy chain (MHC, about 200 kDa), paramyosin (PM, 100–135 kDa), Actin (35–48 kDa), and myosin light chain (MLC, 11–17 kDa) were detected in each group. MHC and Actin, as major proteins in muscle, could influence the texture and water-holding capacity of meat products [46]. The intensity of MHC and Actin bands in the OX group increased compared to the blank group. This result was due to the oxidation of MP, resulting in the formation of protein aggregates [45]. In the FSE-L and FSE-M groups, slight decreases in the intensity of MHC and Actin bands were determined compared with the OX group. This might be related to the formation of disulfide bonds via the components in FSE. The disulfide bonds could be cross-linked with proteins, leading to a decrease in MHC and Actin in the FSE-L and FSE-M groups [45]. This result was in accord with the result of total sulfhydryl content. The decreases in the intensity of MHC and Actin bands in the FSE-L and FSE-M groups indicated that the 0.67% and 2.67% dosages could retard the oxidation of the MP and improve the tenderness and water-holding competence of the duck meat. However, the intensity of MHC and Actin bands in the FSE-H group were darked and compared with the OX group. This result might be that the high dose of FSE aggravated the oxidation of the MP. PM and MLC bands had no distinct changes in each group. Other scholars have found that the pomegranate peel extract treatment could decrease the intensity of protein bands in the MP and prevent protein oxidation after storage for 6 months [18]. 

### 3.3. The Effect of FSE on MP Structure

#### 3.3.1. Secondary Structure

The amide I band in FTIR spectrometry could reflect the conformation changes in the secondary structure. The secondary structure content of the MP was calculated using the Fourier transform deconvolution and second derivative fitting of the amide I band. Figure 4a shows the changes in the secondary structure of the MP in different groups. In the OX group, the content of α-helix was significantly increased than in the blank group (*p* < 0.05). The β-turn content of the MP in the OX group significantly decreased, and the random coil increased significantly compared with the blank group (*p* < 0.05). The results indicated that the Fenton system induced the reorganization of protein structures. In the FSE groups, the α-helix, β-sheet, and random coil content of the MP decreased significantly, and the β-turn content of the MP was significantly increased compared with the OX group (*p* < 0.05). The loss of the β-sheet indicated that FSE might interact with protein molecules, resulting in the disordered and loose secondary structure [47]. The hydrogen bonds were the main force to maintain the α-helix structure of the protein [48]. And the decrease in α-helix content might be due to the fact that FSE disrupted hydrogen bonds among the MP. The modification of MP via FSE led to the transformation of α-helix into a β-turn structure simultaneously. A study also found that the addition of gallic acid, tannic acid, (-)-epigallocatechin gallate, and epigallocatechin decreased the α-helix content and lowered surface hydrophobicity of the pork MP [49]. 

#### 3.3.2. Tertiary Structure

The fluorescence intensity of endogenous tryptophan (Trp) is one of the main indexes to evaluate the tertiary structure of a protein. When the protein was folded, the tryptophan was embedded in the protein and exhibited high fluorescence intensity [50]. As shown in Figure 4b, the fluorescence intensity of the OX group was lower than the blank group, with a redshift from 336 nm to 334 nm. It indicated that there was a decrease in hydrophobicity and an increase in polarity of the oxidative MP. With the addition of FSE, the fluorescence intensity had a further decrease, demonstrating that the addition of FSE could increase the hydrophobicity of the MP. Since the oxidation induced the breakdown of a tertiary structure, the FSE might interact with the exposed tryptophan and tyrosine residues, and the fluorescence of amino acid residues was partly shielded [51]. Additionally, the polysaccharide in FSE might improve the polarity of the MP solution and lead to a decline in fluorescence intensity [52]. A recent study also found that soybean-soluble polysaccharides contributed to the reduction in fluorescence intensity of soybean isolate protein fibers, producing a red shift. At the same time, its surface hydrophobicity was significantly reduced [53].

#### 3.3.3. Particle Size

The particle size distribution of protein could reflect the cross-linking and polymerization of protein. And the cross-linking degree of the protein polypeptide chain could reflect the tenderness of meat products [54]. Figure 5a shows that the particle size distribution of the MP was mainly distributed from 400 nm to 2000 nm. The particle size distribution range of the MP in the OX group was increased compared to the blank group, indicating that the MP in the OX group was aggregated and the structure of the MP gel was destroyed. A previous study found that the particle size of the pork MP increased with the increase in oxidation level [9]. The addition of FSE (0.67%, 2.67%, and 4.67%) could decrease the particle size of the MP compared with the OX group. These results indicated that FSE could inhibit the aggregation and oxidation of the MP and reduce the degree of protein cross-linking, which was consistent with the results of total sulfhydryl content. Compared with the FSE-L group, the particle size of the MP increased from 558 nm to 992 nm with the increase in FSE content, the particle size of FSE-H had a red shift with the improvement in FSE content. It might be attributed to the fact that excessive FSE promoted the aggregation and oxidation of the MP.

#### 3.3.4. Zeta Potential

Zeta potential is a critical index to evaluate the stability of myofibrillar proteins; it reflects the driving force of electrostatic interactions between proteins [55]. Compared with the blank group, Figure 5b shows that the zeta potential of the MP in the OX group significantly increased (*p* < 0.05), indicating a decrease in MP solubility in the OX group. The OX group had a smaller absolute value of potential than the blank group. The increase in the zeta potential of the MP in the OX group might be due to the partial denaturation and aggregate formation of the MP [56]. The addition of FSE decreased the zeta potential of the MP compared with the OX group (*p* < 0.05). And with the increase in the FSE concentration, the absolute value of potential was increased. FSE could urge the unfoldment of the MP and subsequently expose more hydrophobic residues. The hydrophobic residues with like charges repelled and kept the stability of the MP.

#### 3.3.5. Hydrophobicity

The surface hydrophobicity of protein is normally considered to evaluate the extent of protein oxidation and aggregation [57]. In Figure 6a, the amount of bound BPB in the blank group was 22.97 μg, and that of the OX group increased by 18.89%. This might be due to the MP oxidation induced via the Fenton oxidation system, which led to the uncovering of hydrophobic residues and the enhancement of the hydrophobicity of the MP. The amount of bound BPB in the FSE-L group was significantly decreased than that in the blank group and OX group (*p* < 0.05). The results indicated that FSE-L (0.67%) could decrease the hydrophobicity of the MP. The reason might be the formation of complexes between active components in FSE and hydrophobic side chains of amino acids [58]. Additionally, this interaction might induce hydrophilic groups to delay the denaturation of proteins. However, the amount of bound BPB was significantly increased in the FSE-H group than in the OX group (*p* < 0.05). The increase in hydrophobicity in the FSE-H group might be attributed to the high dose of FSE, which promoted the unfoldment of the MP and exposed the internal hydrophobic amino acids. Other research scholars had come up with similar studies that showed that 12 and 60 µM/g of rosmarinic acid decreased the surface hydrophobicity of the MP than oxidized the MP [59]. 

#### 3.3.6. Solubility

Solubility can indicate the level of denaturation of the protein, and the solubility is normally connected to the hydrophobicity of the protein [60]. In Figure 6b, the solubility of the MP decreased from 71.35% to 49.39% after oxidation. The solubility of the MP with the addition of FSE was 67.96% (FSE-L group), 64.30% (FSE-M group), and 56.74% (FSE-H group), respectively. FSE significantly improved the solubility of the MP than the OX group (*p* < 0.05), and it showed a dose-dependent effect in the decrease in MP solubility. FSE could shield the hydrophobic binding site and improve the solubility of oxidative MP. However, the solubility of the FSE-H group was lower than that of the FSE-L group. This result indicated that excessive addition of FSE might combine with the MP and subsequently reduce the size of hydrophilic groups of proteins, leading to the aggregation and decreased solubility of the MP [61]. A previous study reported a similar result that a certain concentration of pomegranate peel extract could improve the total protein solubility and retard protein oxidation in meatballs containing antioxidants after six months of frozen storage [18]. 

#### 3.3.7. Microstructure

A scanning electron microscopy (SEM) investigation was carried out to detect the variations in the MP structure [62]. In Figure 7, the MP in the OX group had a decrease in smoothness and an increase in holes in the structures compared with the blank group, and some small particles were formed in the OX group. It was reported that the formation of a rough structure might be due to the poor solubility induced by the unfolding of the MP [63]. In these three FSE groups, the addition of FSE might combine with the MP, leading to the aggregation and the number of holes decreasing, and a more dense structure occurred compared with the OX group. A previous study also found a similar phenomenon where ginger extract (0.5 mg/mL), hesperidin (0.5 mg/mL), and yam saponin (1 mg/mL) increased the density of gels and protected the oxidation of the myofibrillar protein in different degrees compared with the control [16].

## 4. Conclusions

In this study, the effect of FSE on the MP oxidation induced via the Fenton oxidation system was investigated. The addition of FSE (0.67% and 2.67%) effectively inhibited the accumulation of carbonyl and the loss of total sulfhydryl of the MP. The changes in carbonyl groups and total sulfhydryl indicated that FSE alleviated the oxidative damage of the MP caused by the Fenton system. The increase in MP hydrophobicity and solubility caused by FSE elucidated that FSE could improve the stability and physical properties of the MP. Meanwhile, the addition of FSE decreased the negative surface charge and fluorescence intensity. These results suggested that the interaction between FSE and the MP facilitated the unfolding and disorder of the MP structure. And FSE contributed to the transformation of the α-helix structure to the β-turn structure. SDS-PAGE analysis found that FSE could prevent the aggregation of the MP. SEM observed that FSE decreased the holes in the surface of the MP and facilitated a denser structure. Consequently, FSE has outstanding potential to be a natural antioxidant in the meat products industry. And in future research, the analysis of active ingredients in FSE could be studied based on metabonomics.

## Figures and Tables

**Figure 1 foods-12-04482-f001:**
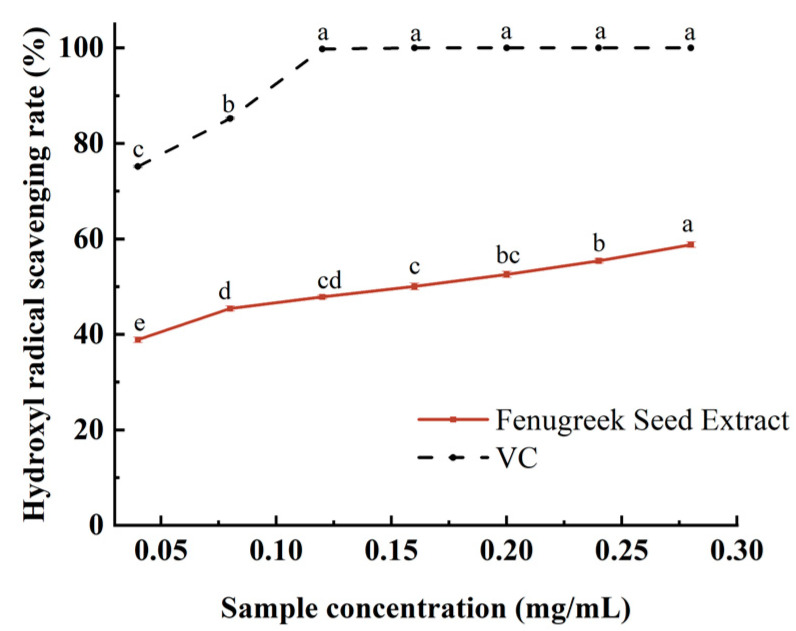
Free radical scavenging ability of FSE with different concentrations (a, b, c etc. represent the significance between groups).

**Figure 2 foods-12-04482-f002:**
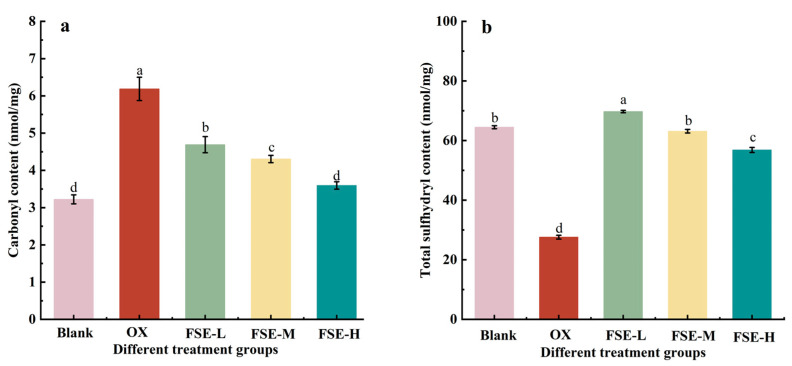
Effects of different concentrations of FSE on MP carbonyl (**a**) and total sulfhydryl group contents (**b**). Values with different letters mean statistically significant (*p* < 0.05).

**Figure 3 foods-12-04482-f003:**
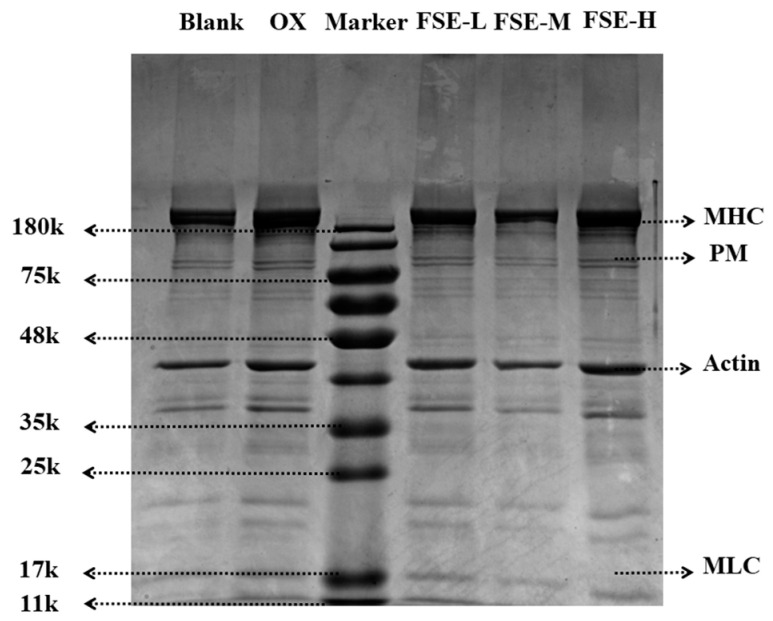
Effects of different concentrations of FSE on the MP SDS-PAGE electrophoretic pattern.

**Figure 4 foods-12-04482-f004:**
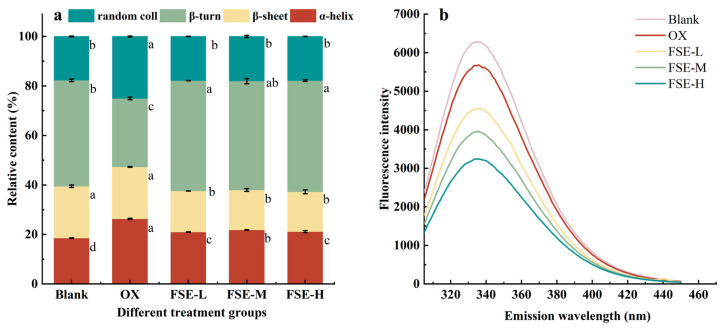
Effects of different concentrations of FSE on the MP secondary structure (**a**) and endogenous tryptophan fluorescence (**b**). Values with different letters mean statistically significant (*p* < 0.05).

**Figure 5 foods-12-04482-f005:**
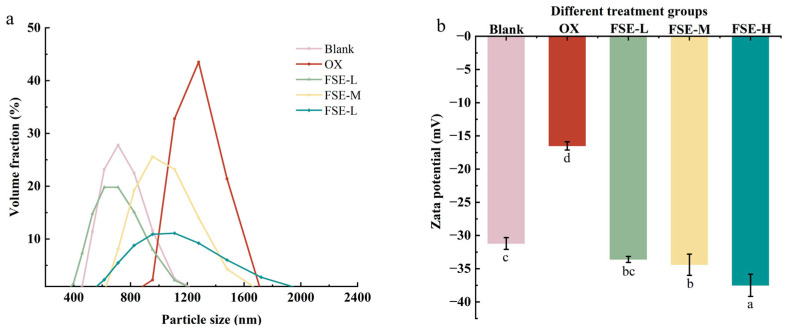
MP particle size distribution in different treatment groups (**a**). Zeta potential of the MP in different treatment groups (**b**). Values with different letters mean statistically significant (*p* < 0.05).

**Figure 6 foods-12-04482-f006:**
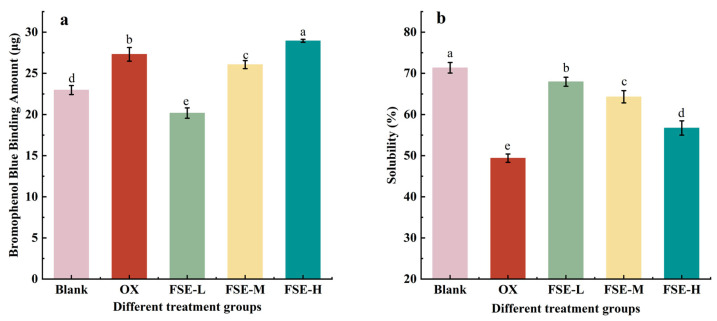
Effects of different concentrations of FSE on the hydrophobicity of the MP surface (**a**) and solubility (**b**). Values with different letters mean statistically significant (*p* < 0.05).

**Figure 7 foods-12-04482-f007:**
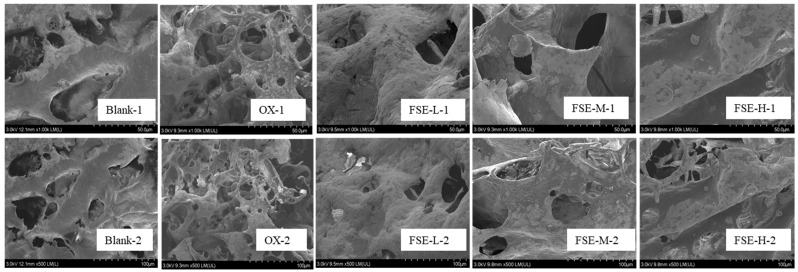
Effect of different concentrations of FSE on the microstructure of the MP. The graphs in the first line were collected at 1000× magnification, and the second line graphs were collected at 500× magnification.

## Data Availability

The data presented in this study are available on request from the corresponding author. The data are not publicly available because that the paper on this part of the data is being submitted.

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
