# Peer review of "Effect of Fenugreek (Trigonella foenum-graecum L.) Seed Extracts on the Structure of Myofibrillar Protein Oxidation in Duck Meat"

_foods, 2023, doi:10.3390/foods12244482_

Round 1

Reviewer 1 Report

Comments and Suggestions for Authors

Title: Effect of Fenugreek (Trigonella foenum-graecum L.) seed extracts on the structure of myofibrillar protein oxidation in duck meat

The manuscript “Effect of Fenugreek (Trigonella foenum-graecum L.) seed extracts on the structure of myofibrillar protein oxidation in duck meat” investigated the effects of fenugreek (Trigonella foenum-graecum L.) seed extracts (FS) on the structure of duck myofibrillar protein (MP) oxidation by Fourier transform infrared (FTIR), fluorescence spectroscopy, SDS-PAGE and scanning electron microscopy (SEM) in the Fenton oxidation system. Study concluded that, FS had positive effects on the antioxidation and could be considered as a natural antioxidant in meat products. It is well written article with some interesting findings; however, there are some places in the manuscript that need to be revised

Line 20: Authors should add part of materials and methods in the abstract part that would be easy for the readers to understand this portion.

Line 23: Remove the word “both”.

Line 34-53: Authors should highlight the importance of duck meat in China may be by giving the example of Beijing roasted duck.

Line 78: Why the drumsticks were purchased from the local market i.e., Huaying Food Co. (Shandong, China); because its oxidation study and there may be oxidation in the samples prior to the treatment, therefore, why authors didn’t perform the slaughtering of the ducks under control environment? In such a way, it would be easy for the scientists to follow the current study? The postmortem time at which treatment is given is unknow? Authors must explain the reason in the manuscript.

Line 90: If the peeled duck leg meat was frozen then authors should mention it at line 78, to make it clear.

Line 97: Remove the word “freezing”.

Line 98: What was the pH of the buffer solution (50 mmol/L KCl, 5 mmol/L β-mercaptoethanol) used for washing?

Line 99: pH of the buffer solution (50 mmol/L potassium chloride, 5 mmol/L β-mercaptoethanol) is missing?

Line 105: Use colon (:) after “using the following procedures”.

Line 104: Is there any reference about the extraction of fenugreek seed extracts?

Line 126-128: Rephrase the sentence.

Line 157: How the mass concentration of MP solution was adjusted to 1 mg/mL?

Line 158-159: How the 5% concentrate gel and 12% separator gel were prepared?

Line 156: Write the name of company of which apparatus was used for SDS-PAGE electrophoresis?

Line 163: How images were recorded? Name of instrument used?

Line 220: Which type of one-way analysis of variance (ANOVA) was used? Explain the method.

How the groups FS-L, FS-M and FS-H were prepared? Authors should mention it in materials and methods part of the manuscript and even in each figure caption. And what was in the blank sample?

Figure 3: Is the identification of the MLC on the SDS-PAGE rightly identified?

Figure 3: It is better to put the marker in the first well for future studies.

Line 268: How the intensities of the protein bands were identified? Did author used any software to identify them? It would be better to use the software such as Quantity 1.0 software to make it clear about the intensities, otherwise, how authors can write the results or discussion based on visual identification?

Line 346: Why the zeta potential values of the FS-M and FS-H groups were lower than the blank group, though, these samples are also containing FS, that author is claiming a good antioxidant?

Authors should discuss the chances of practical implication of the FS regarding industry point of view.

English grammar and sentence structure should be revised and corrected throughout the manuscript. Therefore, minor editing of English language is required.

Comments on the Quality of English Language

English grammar and sentence structure should be revised and corrected throughout the manuscript. Therefore, minor editing of the English language is required.

Author Response

Comments 1: Authors should add part of materials and methods in the abstract part that would be easy for the readers to understand this portion.

Response 1: Thank you for pointing this out. We agree with this comment. Therefore, we have been revised in the manuscript (line 18).

Comments 2: Remove the word “both”.

Response 2: Thank you for pointing this out. We agree with this comment. Therefore, we have been removed in the manuscript (line 22).

Comments 3: Authors should highlight the importance of duck meat in China may be by giving the example of Beijing roasted duck.

Response 3: Thank you for pointing this out. We agree with this comment. Therefore, we have been added in the manuscript (line 38-39).

Comments 4: Why the drumsticks were purchased from the local market i.e., Huaying Food Co. (Shandong, China); because its oxidation study and there may be oxidation in the samples prior to the treatment, therefore, why authors didn’t perform the slaughtering of the ducks under control environment? In such a way, it would be easy for the scientists to follow the current study? The postmortem time at which treatment is given is unknow? Authors must explain the reason in the manuscript.

Response 4: Thank you for pointing this out. The raw material for the experiment is frozen duck legs purchased from Huaying Foods Co. Frozen duck legs are always kept at -20°C before treatment and then uniformly pre-treated, this process will inhibit oxidation of duck legs.

Comments 5: If the peeled duck leg meat was frozen then authors should mention it at line 78, to make it clear.

Response 5: [Peeled duck drumsticks were purchased from Huaying Food Co. (Shandong, China).] Thank you for pointing this out. We agree with this comment. Therefore, we have been revised in the manuscript (line 83).

Comments 6: Remove the word “freezing”.

Response 6: Thank you for pointing this out. We agree with this comment. Therefore, we have been removed in the manuscript (line 105).

Comments 7: What was the pH of the buffer solution (50 mmol/L KCl, 5 mmol/L β-mercaptoethanol) used for washing?

Response 7: Thank you for pointing this out. We agree with this comment. Therefore, we have been revised in the manuscript (line 106).

Comments 8: pH of the buffer solution (50 mmol/L potassium chloride, 5 mmol/L βmercaptoethanol) is missing?

Response 8: Thank you for pointing this out. We agree with this comment. Therefore, we have been revised in the manuscript (line 108).

Comments 9: Use colon (:) after “using the following procedures”.

Response 9: Thank you for pointing this out. We agree with this comment. Therefore, we have been revised in the manuscript (line 113).

Comments 10: Is there any reference about the extraction of fenugreek seed extracts?

Response 10: Thank you for pointing this out. Research was conducted in this lab to derive effect of FSE addition on the tenderness of duck meat, but the paper is in progress and has not yet been published. We agree with this comment. Therefore, we have been revised in the manuscript (line 120).

Comments 11: Rephrase the sentence.

Response 11: Thank you for pointing this out. We agree with this comment. Therefore, we have been revised in the manuscript (line 135-137).

Comments 12: How the mass concentration of MP solution was adjusted to 1 mg/mL

Response 12: Thank you for pointing this out. First, the concentration of the MP solution was 1mg/mL, which was the mass-volume concentration. This meant that 1mL of the solution contained 1mg of MP by mass. Then, depending on the volume of the solution for the different groups, the corresponding mass of MP was added.

Comments 13: How the 5% concentrate gel and 12% separator gel were prepared?

Response 13: Thank you for pointing this out. The 5% concentrate gel was configured from 6.0mL of 30% acrylamide, 3.8mL of 1.5moL/Ltris-HCl, 150uL of 10% SDS, 150uL of 10% ammonium persulfate, 6uL of TEMED, and 4.9mL of distilled water. The 12% separation gel was configured from 1.3 mL of 30% acrylamide, 1.0 mL of 1.5 moL/Ltris-HCl, 80 uL of 10% SDS, 80 uL of 10% ammonium persulfate, 8 uL of TEMED, and 5.5 mL of distilled water.

Comments 14: Write the name of company of which apparatus was used for SDS-PAGE electrophoresis?

Response 14: Thank you for pointing this out. We agree with this comment. Therefore, we have been added in the manuscript (line 169-171).

Comments 15: How images were recorded? Name of instrument used?

Response 15: Thank you for pointing this out. We agree with this comment. Therefore, we have been revised in the manuscript (line 169-171). In an electric field, proteins move toward an electrode with an opposite charge. Proteins come in a variety of sizes and shapes and have charges that correlate with their amino acid composition. Different proteins have their characteristic migration rates. Applying an electric field between the buffer chambers forces the proteins to migrate into the gel and swim in the gel. By visually tracking the movement of proteins through the gel by adding dye to the sample, a downward movement of the dye through the gel can be seen over time. SDS-PAGE gel electrophoresis visualizes the oxidation of amino acid residues containing nucleophilic groups and peptide chain breaks that produce protein fragments or polymers of protein.

Comments 16: Which type of one-way analysis of variance (ANOVA) was used? Explain the method. How the groups FS-L, FS-M and FS-H were prepared? Authors should mention it in materials and methods part of the manuscript and even in each figure caption. And what was in the blank sample?

Response 16: Thank you for pointing this out. Experimental results were analyzed by IBM SPSS 26 with one-way analysis of variance (ANOVA) and Duncan's multiple range test.

The protein solution with 0.67%, 2.67% and 4.67% (w/w) of FSE concentration was used as the FSE-L, FSE-M and FSE-H of the experimental group, respectively. “w/w” means percentage of mass, or it can be said to be percentage of weight. The concentration of a solution expressed as a percentage of the mass of the solute over the total mass of the solution is called the mass percent concentration and is expressed by the symbol “%”. The protein solution of 0.67 %, 2.67 % and 4.67 % ( weight percentage ) FSE refers to the 100 g protein solution containing 0.67 g, 2.67 g and 4.67 g of FSE, respectively.

The protein solution with the Fenton oxidation system was used as the oxidation group (OX), while the protein solution containing EDTA-2Na served as the blank control (Blank).

Comments 17: Is the identification of the MLC on the SDS-PAGE rightly identified?

Response 17: Thank you for pointing this out. We are quite certain that the identification of the MLC on the SDS-PAGE is rightly identified.

Comments 18: It is better to put the marker in the first well for future studies

Response 18: Thank you for pointing this out. We agree with this comment. It is very unfortunate that it is not possible to make changes at this time. In future experiments, we will be sure to put the marker in the first well for future studies.

Comments 19: How the intensities of the protein bands were identified? Did author used any software to identify them? It would be better to use the software such as Quantity 1.0 software to make it clear about the intensities, otherwise, how authors can write the results or discussion based on visual identification?

Response 19: Thank you for pointing this out. According to your requirements, the rest of the electropherograms could not be normalized due to photography problems. The figure 3 of manuscript has been normalized the grayscale value of the band by Image software, and then analyze it more reliably in Table 1. The results showed that compared with the OX group, the degree of MHC protein aggregation in the FSE-L group decreased by 19.87 %, and the degree of Actin protein aggregation decreased by 15.39 %. The degree of MHC protein aggregation in the FSE-M group was reduced by 23.69 %, and the degree of Actin protein aggregation was reduced by 27.65 %. The degree of MHC protein aggregation in FSE-H group decreased by 13.03 %, and the degree of Actin protein aggregation increased by 6.91 %. The increase in the degree of Actin protein aggregation may be due to the high concentration of FSE, which exacerbates the oxidation of MP by reacting too many disulfide bonds with other substances. 2.67% FSE addition has the best effect in delaying the oxidation of MP, indicating that the reaction between protein and disulfide bonds has reached its peak.

Table 1 The result of the grayscale value of the band by Image software

The group

Electrophoresis channel

Area

Mean

InDen

RawlintDen

OX

Background

16200

4.567901235

74000

74000

MHC protein

16200

41.72901235

676010

676010

Actin protein

16200

35.59259259

576600

576600

FSE-L

Background

16200

8.770555556

142083

142083

MHC protein

16200

38.54771605

624473

624473

Actin protein

16200

35.01962963

567318

567318

FSE-M

Background

16200

4.321049383

70001

70001

MHC protein

16200

32.67969136

529411

529411

Actin protein

16200

26.76709877

433627

433627

FSE-H

Background

16200

6.311358025

102244

102244

MHC protein

16200

38.63055556

625815

625815

Actin protein

16200

39.48055556

639585

639585

Comments 20: Why the zeta potential values of the FS-M and FS-H groups were lower than the blank group, though, these samples are also containing FS, that author is claiming a good antioxidant? Authors should discuss the chances of practical implication of the FS regarding industry point of view.

Response 20: Thank you for pointing this out. the zeta potential values of the FSE-M and FSE-H groups were lower than the blank group. It may be because of the addition of FSE. This phenomenon suggested that FSE promoted the unfolding of MP, thereby exposing more hydrophobic residues than the blank group.

Fenugreek seed is an important cash crop in China, and as a medicinal and food plant, it is rich in flavonoids, polysaccharides, coumarins and other antioxidant actives, in addition to fenugreek alkaloid components, which are the main source of its caramel flavor and cigarette aroma. Fenugreek seeds are widely used as a high-quality flavoring in the UK, USA, India and other countries. Therefore, in order to develop the potential of fenugreek seeds for application in food processing in China and to increase the added value of fenugreek plant.

Comments 21: English grammar and sentence structure should be revised and corrected throughout the manuscript. Therefore, minor editing of English language is required

Response 21: Thank you for pointing this out. We agree with this comment. Therefore, the revisions had been marked red in the manuscript.

Reviewer 2 Report

Comments and Suggestions for Authors

Dear authors, the revised manuscript is novel and interesting. According to the review that was carried out, it is necessary to address the following:

Line 17: It is suggested to use the abbreviation FSE instead of FS

Line 38: use 55%–60% instead of 55%-60%

Line 44: use [6–8] instead of [6-8]

Line 54: When the word inhibit is used, it can refer to a total reduction of oxidation, therefore, other terms such as reduce, diminish, delay, prevent, etc. are recommended.

Line 55-57: In addition to mentioning the most common antioxidants, you could include the recommended concentrations for the use of both.

Line 59: Is hesperidin considered a plant?

Line 61: use [16–19] instead of [16-19]

Line 72-74: The title does not reflect the purpose of the indicated study, there must be coherence, the sensory part and/or thermal process is not included in the title.

Line 76-86: Review this section, there are reagents that are mentioned and do not appear in the document. Check that all reagents used appear in this section

Line 79: rewrite… Reagents

Line 87: rewrite… Extraction of Duck Meat MP

Line 90: rewrite… 4 °C for 4 h

Line 90: rewrite… 10,000

Line 90: rewrite… 10 sec

Line 97: insert information (model, trademark, country) of the use equipment (centrifuge)

Line 100: rewrite… 10,000

Line 100: rewrite… 10 sec

Line 103: rewrite… 2.3. Extraction of FS and Construction of Fenton System

Line 106: rewrite… 65 °C for 8 h

Line 106: How was the drying carried out? on stove? dehydrator? indicate equipment information

Line 107: rewrite… 50 °C for 8 h

Line 109: 55 °C

Line 110: 72 h

Line 114: insert space… solution [27].

Line 122: 2.4. Hydroxyl Radicals Scavenging Capacity

Line 132: It is necessary to include the equation number in all those used in the document

Line 133: 2.5. Determination of Carbonyl Content

Line 137: 25 °C

Line 139: r or rpm?

Line 139: indicate temperature condition

Line 142: 37 °C

Line 143: indicate temperature condition

Line 145: 22,000

Line 146: 2.6. Determination of Total Sulfhydryl Content

Line 149: Regarding the centrifugation conditions of the samples, sometimes rpm is used and other times g, it is necessary to express them in a single type of unit.

Line 154: 13,600

Line 155: Electrophoresis

Line 173: 2.9. Fluorescence Spectroscopy of Endogenous Tryptophan

Line 180: 2.10. Particle Size

Line 183: 10,000

Line 183: 4 °C

Line 185: 25 °C

Line 187: 2.11. Zeta Potential

Line 189: 10,000

Line 190: 4 °C

Line 192: 25 °C

Line 193: 2.12. Determination of Surface Hydrophobicity

Line 198: 25 °C

Line 207: 12,000

Line 207: insert used temperature conditions

Line 213: 2.14. Scanning Electron Microscope Analysis

Line 218: 2.15. Statistical Analysis

Line 223: Discussion

Line 224: 3.1. Antioxidant Capacity of FSE

Line 225: Explain why only the hydroxyl radical scavenging activity test was used.

Line 231: insert space… rate (%)

Line 233: 3.2. The Effect of FS on MP Oxidation

Line 234: 3.2.1. Carbonyl and total sulfhydryl contents in MP

Line 234: Indicate in this section the limit value for the total formation of carbonyls and sulfhydryl to establish that a meat product is not suitable for consumption.

Line 237: use [41–43] instead of [41-43]

Line 246: scientific names must be written in italic text format

Line 271: rewrite… water holding

Line 289: 3.3. The Effect of FS on MP Structure

Line 343: 3.3.4. Zeta potential

Line 355: insert space… fraction (%)

Line 438: According to the authors' guide, the journal of each reference must be abbreviated. Furthermore, in page numbering it is necessary to use – instead of -.

Line 462: standardize titles, do not use capital letters in the first letter of each word (review through this section)

Author Response

Comments 1: It is suggested to use the abbreviation FSE instead of FS

Response 1: Thank you for pointing this out. We agree with this comment. Therefore, we have been revised in the manuscript and figures.

Comments 2: use 55%–60% instead of 55%-60%

Response 2: Thank you for pointing this out. We agree with this comment. Therefore, we have been revised in the manuscript (line 41).

Comments 3: use [6–8] instead of [6-8]

Response 3: Thank you for pointing this out. We agree with this comment. Therefore, we have been revised in the manuscript (line 47).

Comments 4: When the word inhibit is used, it can refer to a total reduction of oxidation, therefore, other terms such as reduce, diminish, delay, prevent, etc. are recommended.

Response 4: Thank you for pointing this out. We agree with this comment. Therefore, we have been revised in the manuscript (line 57 ).

Comments 5: In addition to mentioning the most common antioxidants, you could include the recommended concentrations for the use of both.

Response 5: Thank you for pointing this out. We agree with this comment. Therefore, we have been revised in the manuscript (line 59-60).

Comments 6: Is hesperidin considered a plant?

Response 6: Thank you for pointing this out. We agree with this comment. Therefore, we have been revised in the manuscript (line 63) because of hesperidin is a flavonoid widely found in citrus fruits.

Comments 7: use [16–19] instead of [16-19]

Response 7: Thank you for pointing this out. We agree with this comment. Therefore, we have been revised in the manuscript (line 65).

Comments 8: The title does not reflect the purpose of the indicated study, there must be coherence, the sensory part and/or thermal process is not included in the title.

Response 8: Thank you for pointing this out. We agree with this comment. Therefore, we have been revised in the manuscript (line 76-79).

Comments 9: Review this section, there are reagents that are mentioned and do not appear in the document. Check that all reagents used appear in this section

Response 9: Thank you for pointing this out. We agree with this comment. Therefore, we have been revised in the manuscript (line 85-93).

Comments 10: rewrite… Reagents

Response 10: Thank you for pointing this out. We agree with this comment. Therefore, we have been revised in the manuscript (line 81).

Comments 11: rewrite… Extraction of Duck Meat MP

Response 11: Thank you for pointing this out. We agree with this comment. Therefore, we have been revised in the manuscript (line 95).

Comments 12: rewrite…4 °C for 4 h

Response 12: Thank you for pointing this out. We agree with this comment. Therefore, we have been revised in the manuscript (line 98).

Comments 13: rewrite… 10,000

Response 13: Thank you for pointing this out. We agree with this comment. Therefore, we have been revised in the manuscript (line 103).

Comments 14: rewrite… 10 sec

Response 14: Thank you for pointing this out. We agree with this comment. Therefore, we have been revised in the manuscript (line 103).

Comments 15: rewrite… 2.3. Extraction of FS and Construction of Fenton System

Response 15: Thank you for pointing this out. We agree with this comment. Therefore, we have been revised in the manuscript (line 111 ).

Comments 16: rewrite… 65°C for 8 h

Response 16: Thank you for pointing this out. We agree with this comment. Therefore, we have been revised in the manuscript (line 114 ).

Comments 17: How was the drying carried out? on stove? dehydrator? indicate equipment information

Response 17: Thank you for pointing this out. We agree with this comment. Therefore, we have been added in the manuscript (line 114).

Comments 18: rewrite… 50°C for 8 h

Response 18: Thank you for pointing this out. We agree with this comment. Therefore, we have been revised in the manuscript (line 116).

Comments 19: 55 °C

Response 19: Thank you for pointing this out. We agree with this comment. Therefore, we have been revised in the manuscript (line 118).

Comments 20: 72 h

Response 20: Thank you for pointing this out. We agree with this comment. Therefore, we have been revised in the manuscript (line 119).

Comments 21: insert space… solution [27].

Response 21: Thank you for pointing this out. We agree with this comment. Therefore, we have been added in the manuscript (line 123).

Comments 22: 2.4. Hydroxyl Radicals Scavenging Capacity

Response 22: Thank you for pointing this out. We agree with this comment. Therefore, we have been revised in the manuscript (line 131).

Comments 23: It is necessary to include the equation number in all those used in the document

Response 23: Thank you for pointing this out. We agree with this comment. Therefore, we have been revised in the manuscript (line 141, 212, 220).

Comments 24: 2.5. Determination of Carbonyl Content

Response 24: Thank you for pointing this out. We agree with this comment. Therefore, we have been revised in the manuscript (line 142).

Comments 25: 25 °C

Response 25: Thank you for pointing this out. We agree with this comment. Therefore, we have been revised in the manuscript (line 145 ).

Comments 26: r or rpm?

Response 26: Thank you for pointing this out. We agree with this comment. Therefore, we have been revised in the manuscript (line 147).

Comments 27: indicate temperature condition

Response 27: Thank you for pointing this out. We agree with this comment. Therefore, we have been added in the manuscript (line 148).

Comments 28: 37 °C

Response 28: Thank you for pointing this out. We agree with this comment. Therefore, we have been revised in the manuscript (line 151).

Comments 29: indicate temperature condition

Response 29: Thank you for pointing this out. We agree with this comment. Therefore, we have been added in the manuscript (line 152).

Comments 30: 22,000

Response 30: Thank you for pointing this out. We agree with this comment. Therefore, we have been revised in the manuscript (line 153).

Comments 31: 2.6. Determination of Total Sulfhydryl Content

Response 31: Thank you for pointing this out. We agree with this comment. Therefore, we have been revised in the manuscript (line 155).

Comments 32: Regarding the centrifugation conditions of the samples, sometimes rpm is used and other times g, it is necessary to express them in a single type of unit.

Response 32: Thank you for pointing this out. We agree with this comment. Therefore, we have been revised in the manuscript (line 158).

Comments 33: 13,600

Response 33: Thank you for pointing this out. We agree with this comment. Therefore, we have been revised in the manuscript (line 163).

Comments 34: Electrophoresis

Response 34: Thank you for pointing this out. We agree with this comment. Therefore, we have been revised in the manuscript (line 165).

Comments 35: 2.9. Fluorescence Spectroscopy of Endogenous Tryptophan

Response 35: Thank you for pointing this out. We agree with this comment. Therefore, we have been revised in the manuscript (line 184).

Comments 36: 2.10. Particle Size

Response 36: Thank you for pointing this out. We agree with this comment. Therefore, we have been revised in the manuscript (line 191).

Comments 37: 10,000

Response 37: Thank you for pointing this out. We agree with this comment. Therefore, we have been revised in the manuscript (line 194).

Comments 38: 4 °C

Response 38: Thank you for pointing this out. We agree with this comment. Therefore, we have been revised in the manuscript (line 194).

Comments 39: 25 °C

Response 39: Thank you for pointing this out. We agree with this comment. Therefore, we have been revised in the manuscript (line 196).

Comments 40: 2.11. Zeta Potential

Response 40: Thank you for pointing this out. We agree with this comment. Therefore, we have been revised in the manuscript (line 197).

Comments 41: 10,000

Response 41: Thank you for pointing this out. We agree with this comment. Therefore, we have been revised in the manuscript (line 199).

Comments 42: 4 °C

Response 42: Thank you for pointing this out. We agree with this comment. Therefore, we have been revised in the manuscript (line 200).

Comments 43: 25 °C

Response 43: Thank you for pointing this out. We agree with this comment. Therefore, we have been revised in the manuscript (line 202).

Comments 44: 2.12. Determination of Surface Hydrophobicity

Response 44: Thank you for pointing this out. We agree with this comment. Therefore, we have been revised in the manuscript (line 203).

Comments 45: 25 °C

Response 45: Thank you for pointing this out. We agree with this comment. Therefore, we have been revised in the manuscript (line 208).

Comments 46: 12,000

Response 46: Thank you for pointing this out. We agree with this comment. Therefore, we have been revised in the manuscript (line 217).

Comments 47: insert used temperature conditions

Response 47: Thank you for pointing this out. We agree with this comment. Therefore, we have been revised in the manuscript (line 217-218).

Comments 48: 2.14. Scanning Electron Microscope Analysis

Response 48: Thank you for pointing this out. We agree with this comment. Therefore, we have been revised in the manuscript (line 223).

Comments 49: 2.15. Statistical Analysis

Response 49: Thank you for pointing this out. We agree with this comment. Therefore, we have been revised in the manuscript (line 228).

Comments 50: Discussion

Response 50: Thank you for pointing this out. We agree with this comment. Therefore, we have been revised in the manuscript (line 233).

Comments 51: 3.1. Antioxidant Capacity of FSE

Response 51: Thank you for pointing this out. We agree with this comment. Therefore, we have been revised in the manuscript (line 234).

Comments 52: Explain why only the hydroxyl radical scavenging activity test was used.

Response 52: Thank you for pointing this out. Hydroxyl radical scavenging capacity is one of the most important indicators for assessing antioxidant capacity. Substances with high hydroxyl radical scavenging capacity have strong antioxidant ability and can scavenge harmful free radicals in the body, thus protecting cells from oxidative damage. Therefore, the determination of hydroxyl radical scavenging capacity is representative for evaluating the antioxidant properties of substances.

Comments 53: insert space… rate (%)

Response 53: Thank you for pointing this out. We agree with this comment. Therefore, we have been revised in the manuscript (line 241).

Comments 54: 3.2. The Effect of FS on MP Oxidation

Response 54: Thank you for pointing this out. We agree with this comment. Therefore, we have been revised in the manuscript (line 243).

Comments 55: 3.2.1. Carbonyl and total sulfhydryl contents in MP

Response 55: Thank you for pointing this out. We agree with this comment. Therefore, we have been revised in the manuscript (line 244).

Comments 56: Indicate in this section the limit value for the total formation of carbonyls and sulfhydryl to establish that a meat product is not suitable for consumption.

Response 56: Thank you for pointing this out. According to relevant literature, the maximum limit values for carbonyl and sulfhydryl group formation in meat products have not been reported. As the degree of oxidation increases, the more carbonyl groups are formed and the more sulfhydryl groups are lost, which is an infinite value.

Comments 57: use [41–43] instead of [41-43]

Response 57: Thank you for pointing this out. We agree with this comment. Therefore, we have been revised in the manuscript (line 248).

Comments 58: scientific names must be written in italic text format

Response 58: Thank you for pointing this out. We agree with this comment. Therefore, we have been revised in the manuscript (line 256).

Comments 59: rewrite… water holding

Response 59: Thank you for pointing this out. We agree with this comment. Therefore, we have been revised in the manuscript (line 280).

Comments 60: 3.3. The Effect of FS on MP Structure

Response 60: Thank you for pointing this out. We agree with this comment. Therefore, we have been revised in the manuscript (line 299).

Comments 61: 3.3.4. Zeta potential

Response 61: Thank you for pointing this out. We agree with this comment. Therefore, we have been revised in the manuscript (line 353).

Comments 62: insert space… fraction (%)

Response 62: Thank you for pointing this out. We agree with this comment. Therefore, we have been revised in the manuscript (line 365).

Comments 63: According to the authors' guide, the journal of each reference must be abbreviated. Furthermore, in page numbering it is necessary to use – instead of -.

Response 63: Thank you for pointing this out. We agree with this comment. Therefore, we have been revised in the manuscript (line 447-601).

Comments 64: standardize titles, do not use capital letters in the first letter of each word (review through this section)

Response 64: Thank you for pointing this out. We agree with this comment. Therefore, we have been revised and marked red in the manuscript.

Round 2

Reviewer 1 Report

Comments and Suggestions for Authors

The manuscript is sufficiently improved according to the comments and suggestions of the reviewer, therefore, may be accepted for the publication in Foods.

Author Response

Thank you very much for the reviewer 's comments, we will continue to work hard with this affirmation in the future research.